# Model Rubik's Cube: Twisting Resolution, Depth and Width for TinyNets

**Kai Han**[1,2*] **Yunhe Wang**[1*] **Qiulin Zhang**[1,3] **Wei Zhang**[1] **Chunjing Xu**[1] **Tong Zhang**[4]
[1]Noah's Ark Lab, Huawei Technologies
[2]State Key Lab of Computer Science, ISCAS & UCAS
[3]BUPT    [4]HKUST
{kai.han,yunhe.wang,wz.zhang,xuchunjing}@huawei.com, qiulinzhang@bupt.edu.cn
tongzhang@tongzhang-ml.org

## Abstract

To obtain excellent deep neural architectures, a series of techniques are carefully designed in EfficientNets. The giant formula for simultaneously enlarging the resolution, depth and width provides us a Rubik's cube for neural networks. So that we can find networks with high efficiency and excellent performance by twisting the three dimensions. This paper aims to explore the twisting rules for obtaining deep neural networks with minimum model sizes and computational costs. Different from the network enlarging, we observe that resolution and depth are more important than width for tiny networks. Therefore, the original method, *i.e.* the compound scaling in EfficientNet is no longer suitable. To this end, we summarize a tiny formula for downsizing neural architectures through a series of smaller models derived from the EfficientNet-B0 with the FLOPs constraint. Experimental results on the ImageNet benchmark illustrate that our TinyNet performs much better than the smaller version of EfficientNets using the inversed giant formula. For instance, our TinyNet-E achieves a 59.9% Top-1 accuracy with only 24M FLOPs, which is about 1.9% higher than that of the previous best MobileNetV3 with similar computational cost. Code will be available at `https://github.com/huawei-noah/ghostnet/tree/master/tinynet_pytorch`, and `https://gitee.com/mindspore/mindspore/tree/master/model_zoo/research/cv/tinynet`.

## 1 Introduction

Deep convolutional neural networks (CNNs) have achieved great success in many visual tasks, such as image recognition [21, 12, 9], object detection [36, 28, 8], and super-resolution [20, 46, 40]. In the past few decades, the evolution of neural architectures has greatly increased the performance of deep learning models. From LeNet [22] and AlexNet [21] to modern ResNet [12] and EfficientNet [43], there are a number of novel components including shortcuts and depth-wise convolution. Neural architecture search [61, 27, 52, 54, 44] also provides more possibility of network architectures. These various architectures have provided candidates for a large variety of real-world applications.

To deploy the networks on mobile devices, the depth, the width and the image resolution are continuously adjusted to reduce memory and latency. For example, ResNet [12] provides models with different number of layers, and MobileNet [15, 38] changes the number of channels (*i.e.* the width of neural network) and image resolution for different FLOPs. Most of existing works only scale one of the three dimensions – resolution, depth, and width (denoted as $r$, $d$, and $w$). Tan and Le explore the EfficientNet [43], which enlarges CNNs with a compound scaling method. The great

---

success made by EfficientNets bring a Rubik's cube to the deep learning community, *i.e.* we can twist it for better neural architectures using some pre-defined formulas. For example, the EfficientNet-B7 is derivate from the B0 version by uniformly increasing these three dimensions. Nevertheless, the original EfficientNet and some improved versions [2, 50] only discuss the giant formula, the rules for effectively downsize the baseline model has not been fully investigated.

The straightforward way for designing tiny networks is to apply the experience used in Efficient-Net [43]. For example, we can obtain an EfficientNet-B$^{-1}$ with a 200M FLOPs (floating-point operations). Since the giant formula is explored for enlarging networks, this naive strategy could not perfectly find a network with the highest performance. To this end, we randomly generate 100 models by twisting the three dimensions $(r, d, w)$ from the baseline EfficientNet-B0. FLOPs of these models are less than or equal to that of the baseline. It can be found in Figure 1, the performance of best models is about 2.5% higher than that of models obtained using the inversed giant formula of EfficientNet (green line) with different FLOPs.

In this paper, we study the relationship between the accuracy and the three dimensions $(r, d, w)$ and explore a tiny formula for the model Rubik's cube. Firstly, we find that resolution and depth are more important than width for retaining the performance of a smaller neural architecture. We then point out that the inversed giant formula, *i.e.* the compound scaling method in EfficientNets is no longer suitable for designing portable networks for mobile devices, due to the reduction on the resolution is relatively large. Therefore, we explore a tiny formula for the cube through massive experiments and observations. In contrast to the giant formula in EfficientNet that is handcrafted, the proposed scheme twists the three dimensions based on the observation of frontier models. Specifically, for the given upper limit of FLOPs, we calculate the optimal resolution and depth exploiting the tiny formula, *i.e.* the Gaussian process regression

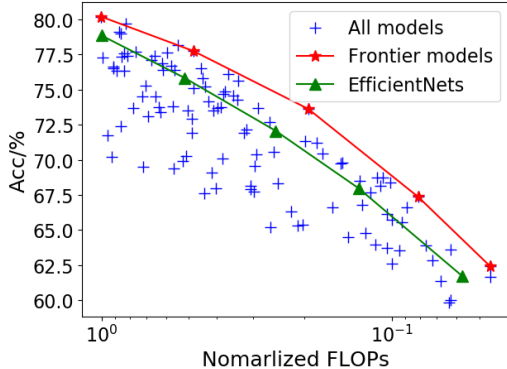

Figure 1: Accuracy *v.s.* FLOPs of the models randomly donwsized from EfficientNet-B0. Five models using the inversed giant formula (green) and the frontier models (red) with both higher performance and lower FLOPs are highlighted.

on frontier models. The width of the resulting model is then determined according to the FLOPs constraint and previously obtained $r$ and $d$. The proposed tiny formula for establishing TinyNets is simple yet effective. For instance, TinyNet-A achieves a 76.8% Top-1 accuracy with about 339M FLOPs but the EfficientNet-B0 with the similar performance needs about 387M FLOPs. In addition, TinyNet-E achieves a 59.9% Top-1 accuracy with only 24M FLOPs, being 1.9% higher than the previous best MobileNetV3 with similar FLOPs. To our best knowledge, we are the first to study how to generate tiny neural networks via simultaneously twisting resolution, depth and width. Besides the validations on EfficientNet, our tiny formula can be directly applied on ResNet architectures to obtain small but effective neural networks.

## 2   Related Work

Here we revisit the existing model compression methods for shrinking neural networks, and discuss about resolution, depth and width of CNNs.

**Model Compression.**   Model compression aims to reduce the computation, energy and storage cost, which can be categorized into four main parts: pruning, low-bit quantization, low-rank factorization and knowledge distillation. Pruning [11, 23, 39, 25, 45, 47, 24] is used to reduce the redundant parameters in neural networks that are insensitive to the model performance. For example, [23] uses $\ell_1$-norm to calculate the importance of each filter and prunes the unimportant ones accordingly. ThiNet [30] prunes filters based on statistics computed from their next layers. Low-bit quantization [17, 60, 35, 29, 7, 10, 18] represents weights or activations in neural networks using low-bit values. DorefaNet [60] trains neural networks with both low-bit weights and activations. BinaryNet [17] and XNORNet [35] quantize each neuron into only 1-bit and learn the binary weights or activations

directly during the model training. Low-rank factorization methods try to estimate the informative parameters using matrix/tensor decomposition [19, 5, 56, 53, 57]. Low-rank factorization achieves some advances in model compression, but it involves complex decomposition operations and is thus computationally expensive. Knowledge distillation [13, 37, 55, 51] attempts to teach a compact model, also called student model, with knowledge distilled from a large teacher network. The common part of these compression methods is that their performance is usually upper bounded by the given pretrained models.

**Resolution, Depth and Width of CNNs.** The three dimensions including resolution, depth and width of convolutional neural networks have much impact on the performance and have been explored for scaling the CNNs. ResNet [12] proposes models of different depth, from ResNet-18 to ResNet-152, to provide choices between model size and model performance. WideResNet [48] propose to decrease the depth and increase the width of residual networks, demonstrating that a wider network is superior to a deep and thin counterpart. The input images of higher resolution provides more information that is helpful to model performance but also leads to higher computation cost [38, 14]. Considering all the three CNN dimensions into account, EffectiveNet [43] proposes a compound scaling method to scale up networks with a handcrafted formula. However, it is still an open problem of how to shrink a given model to small and compact versions.

## 3 Approach

In this section, we first rethink the importance of resolution, depth and width, and find original EfficientNet rule lose its efficiency for smaller models. Based on the observation, we propose a new tiny formula for model Rubik's cube to generate smaller neural networks.

### 3.1 Rethinking the Importance of $(r, d, w)$

Given a baseline CNN, we aim to find the smaller versions of it for deployment on low-resource devices. Resolution, depth and width are three key factors that affect the performance of CNNs as discussed in EfficientNet [43]. However, which of them has more impact on the performance has not been well investigated in the previous works. Here we propose to evaluate the impact of $(r, d, w)$ under the fixed FLOPs or memory constraint. In practice, the FLOPs constraint is more common so we explore under FLOPs constraint and the methods can also be applied for memory constraint. To be specific, the FLOPs of the given baseline CNN are $\mathcal{C}_0$, the resolution of the input image is $\mathcal{R}_0 \times \mathcal{R}_0$, the width is $\mathcal{W}_0$ and the depth is $\mathcal{D}_0$.

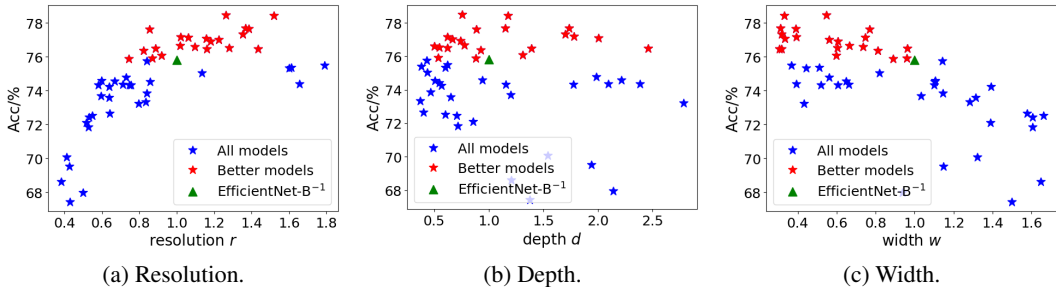

(a) Resolution.  (b) Depth.  (c) Width.

Figure 2: Accuracy *v.s.* $(r, d, w)$ for the models with $\sim$200M FLOPs on ImageNet-100.

**Resolution Is More Important.** Here we start from EfficientNet-B0 with $\mathcal{C}_0$ FLOPs, and sample models with FLOPs of around $0.5\mathcal{C}_0$. In order to search around the target FLOPs, we randomly search the resolution and the depth, and tune the width around $w = \sqrt{0.5/d/r^2}$ to make the resulted model has the target FLOPs (the difference $\leq 3\%$). These random searched models are fully trained for 100 epochs on ImageNet-100 dataset. As shown in Figure 2, the accuracy is more related to the resolution, compared with the depth and the width. We find that the top accuracies are obtained around the range from 0.8 to 1.4. When $r < 0.8$, the accuracy is higher if the resolution is larger, while the accuracy drops slightly when $r > 1.4$. As for the depth, the models with high performance may have various depth from 0.5 to 2, that is to say, we may miss some good models if we narrowly

restrict the depth. When fixing FLOPs, the width has roughly negative correlation to the accuracy. The good models are mostly distributed at $w < 1$.

If we follow the EfficientNet rule to obtain a model with $0.5\mathcal{C}_0$ FLOPs, namely EfficientNet-B$^{-1}$, whose three dimensions are calculated and tuned as $r = 0.86, d = 0.8, w = 0.89$. Its accuracy on ImageNet-100 is only 75.8%, which is far from the optimal combination for $0.5\mathcal{C}_0$ FLOPs. It can be found in Figure 2, there is a number of models with higher performance even though they are randomly generated. This observation motivates us to explore a new model twisting formula that can obtain better models under a fixed FLOPs constraint.

## 3.2 Tiny Formula for Model Rubik's Cube

For a given arbitrary baseline neural network, and with a FLOPs constraint of $c \cdot \mathcal{C}_0$, where $0 < c < 1$ is the reduction factor, our goal is to provide the optimal values of the three dimensions $(r, d, w)$ for shrinking the model. Basically, we assume the optimal coefficients $r$, $w$, $d$ for shrinking resolution, width and depth are

$$r = f_1(c), \quad w = f_2(c), \quad d = f_3(c), \tag{1}$$

where $f_1(\cdot)$, $f_2(\cdot)$ and $f_3(\cdot)$ are the functions for calculating the three dimensions. We will give the formulation of the equations in the following.

Then, we randomly sample a number of models with different coefficients and verify them to explore the relationship between the performance and the three dimensions. The coefficients are randomly sampled from a given range. We preserve the models whose FLOPs are between $0.03 \cdot \mathcal{C}_0$ and $1.05 \cdot \mathcal{C}_0$. After fully training and testing these models, we can obtain their accuracies on validation set. We plot scatter diagram of accuracy *v.s.* FLOPs as shown in Figure 1. Obviously, there are a number of models whose performance is better than the vanilla EfficientNet-B0 and its shrunken versions obtained by exploiting the inversed compound scaling scheme.

To further explore the property of the best models, we select the models on the Accuracy-FLOPs Pareto front. Pareto front is a set of nondominated solutions, being chosen as optimal, if no objective can be improved without sacrificing at least one other objective [32, 3]. In particular, the top 20% models with higher performance and lower computational complexities (*i.e.* FLOPs) are selected using NSGA-III nondominated sorting strategy [3]. We show the relation between depth/width/resolution and FLOPs of these selected models in Figure 3. Spearman correlation coefficient (Spearmanr) are calculated to measure the correlation between depth/width/resolution and FLOPs. From the results in Figure 3, the rank of correlations between the three dimensions with FLOPs is $r > d > w$. Wherein, the Spearmanr score for resolution is 0.81, which is much higher than that of width.

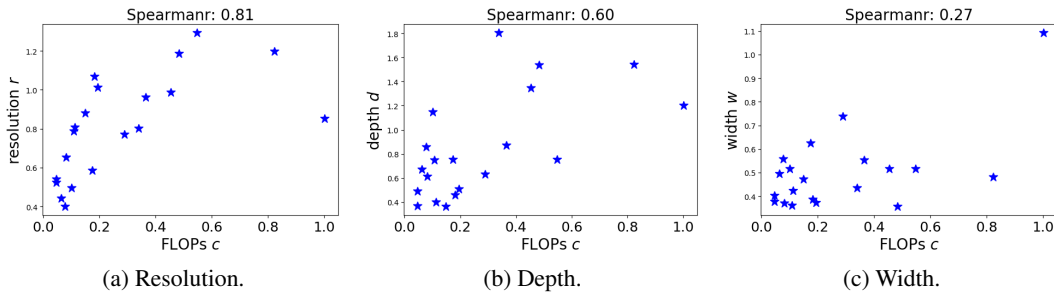

(a) Resolution.      (b) Depth.      (c) Width.

Figure 3: Resolution/depth/width *v.s.* FLOPs for the models on Pareto front.

Inspired by the observation of relationship between $(r, d, w)$ and FLOPs, we propose the tiny formula for shrinking neural architectures as follows. For the given requirement of FLOPs $c \cdot \mathcal{C}_0$, our goal is to calculate the optimal combinations of $(r, d, w)$ for building models with high performance. Since the correlations between resolution/depth and FLOPs are higher than that of width, we present to first twist the resolution and depth to ensure the performance of the smaller model. Taking the formula of resolution as the example, the nonparametric Guassign process regression [34] is utilized to model the mapping from $c \in \mathbb{R}$ to $r \in \mathbb{R}$. Let $\{(c^i, r^i)\}_{i=1}^m$ be the training set with $m$ i.i.d. examples from Figure 3(a), we have

$$r^i = g(c^i) + \epsilon^i, \quad i = 1, \cdots, m, \tag{2}$$

where $\epsilon^i$ is i.i.d. noise variable from $\mathcal{N}(0, \sigma^2)$ distribution, and $g(\cdot)$ follows the zero-mean Gaussian process prior, *i.e.* $f(\cdot) \sim \mathcal{GP}(0, k(\cdot, \cdot))$ with covariance function $k(\cdot, \cdot)$. For convenience, we denote $\vec{c} = [c^1, c^2, \cdots, c^m]^T \in \mathbb{R}^m, \vec{r} = [r^1, r^2, \cdots, r^m] \in \mathbb{R}^m$. Then the joint prior distribution of the training data $\vec{c}$ and the test point $c_*$ belongs to a Gaussian distribution:

$$\begin{bmatrix} \vec{r} \\ r_* \end{bmatrix} \bigg| \vec{c}, c_* \sim \mathcal{N}\left(\vec{0}, \begin{bmatrix} K(\vec{c}, \vec{c}) + \sigma^2 I & K(\vec{c}, c_*) \\ K(c_*, \vec{c}) & k(c_*, c_*) + \sigma^2 \end{bmatrix}\right), \tag{3}$$

where $K(\vec{c}, \vec{c}) \in \mathbb{R}^{m \times m}$ in which $(K(\vec{c}, \vec{c}))_{ij} = k(c^i, c^j)$, $K(\vec{c}, c_*) \in \mathbb{R}^{m \times 1}$, and $K(c_*, \vec{c}) \in \mathbb{R}^{1 \times m}$. Thus, we can calculate the posterior distribution of prediction $r_*$ as

$$r_* | \vec{r}, \vec{c}, c_* \sim \mathcal{N}(\mu_*, \Sigma_*), \tag{4}$$

where $\mu_* = K(c_*, \vec{c})(K(\vec{c}, \vec{c}) + \sigma^2 I)^{-1}\vec{r}$ and $\Sigma_* = k(c_*, c_*) + \sigma^2 - K(c_*, \vec{c})(K(\vec{c}, \vec{c}) + \sigma^2 I)^{-1}K(\vec{c}, c_*)$ are the mean and variance for the test point $c_*$. The formula for depth can be obtained similarly. Then, the last dimension, *i.e.* width, can be determined by FLOPs constraint:

$$w = \sqrt{c/(r^2 d)}, \quad \text{s.t. } 0 < c < 1. \tag{5}$$

In contrast to the handcrafted compound scaling method in EfficientNets, the proposed model shrinking rule is designed based on the observation of frontier small models, which are more effective for producing tiny networks with higher performance.

**TinyNet.** Our tiny formula for model Rubik's cube can be applied to any network architecture. Here we start from the excellent baseline network, EfficientNet-B0 [43], and apply our shrinking method to obtain smaller networks. With about 5.3M parameters and 390M FLOPs, EfficientNet-B0 consists of 16 mobile inverted residual bottlenecks [38], in addition to the normal stem layer and classification head layers. To apply our tiny formula, we first construct a number of networks whose $(r, w, d)$ are randomly sampled as shown in Figure 1. After obtaining the accuracy on ImageNet-100, we can train the Gaussian process regression model for resolution and depth. Here we adopt the widely used RBF kernel as the covariance function. Then, given the desired FLOPs constraint $c$, we can determine the three dimensions, aka $(r, w, d)$, by the above equations. We set $c$ in $\{0.9, 0.5, 0.25, 0.13, 0.06\}$, and obtain a series of smaller EfficientNet-B0, namely, TinyNet-A to E.

## 4 Experiments

In this section, we apply our tiny formula for model Rubik's cube to shrink EfficientNet-B0 and ResNet-50. The effectiveness of our method is verified on the visual recognition benchmarks.

### 4.1 Datasets and Experimental Settings

**ImageNet-1000.** ImageNet ILSVRC2012 dataset [4] is a large-scale image classification dataset containing 1.2 million images for training and 50,000 validation images belonging to 1,000 categories. We use the common data augmentation strategy [41, 14] including random crop, random flip and color jitter. The base input resolution is 224 for $r = 1$.

**ImageNet-100.** ImageNet-100 is the subset of ImageNet-1000 that contains randomly sampled 100 classes. 500 training images are randomly sampled for each class, and the corresponding 5,000 images are used as validation set. The data augmentation strategy is the same as that in ImageNet-1000.

**Implementation details.** All the models are implemented using PyTorch [33] and trained on NVIDIA Tesla V100 GPUs. The EfficientNet-B0 based models are trained using similar settings as [43]. We train the models for 450 epochs using the RMSProp optimizer with momentum 0.9 and decay 0.9. The weight decay is 1e-5 and batch normalization momentum is set as 0.99. The initial learning rate is 0.048 and decays by 0.97 every 2.4 epochs. Learning rate warmup [6] is applied for the first 3 epochs. The batch size is 1024 for 8 GPUs with 128 images per chip. The dropout of 0.2 is applied on the last fully-connected layer for regularization. We also use exponential moving average (EMA) with decay 0.9999. For ResNets, the models are trained for 90 epochs with batch size of 1024. SGD optimizer with the momentum 0.9 and weight decay 1e-4 is used to update the weights. The learning rate starts from 0.4 and decays by 0.1 every 30 epochs.

In the original EfficientNet rule for giant models [43], the FLOPs value of a model is calculated as $2^{-\phi} \cdot \mathcal{C}_0$. We denote the models obtained from the inversed giant formula in original EfficientNet as EfficientNet-B$^{-\phi}$ where $\phi = 1, 2, 3, 4$, with about 200M, 100M, 50M, 25M FLOPs, respectively.

## 4.2 Experiments on ImageNet-100

**Random Sample Results.** As stated in the above sections, we randomly sample a number of models with different resolution, depth and width. In particular, resolution, depth or width is randomly sampled from the range of $0.35 \leq r \leq 2.8$, $0.35 \leq d \leq 2.8$ and $0.35 \leq w \leq 2.8$. The sampled models are trained on ImageNet-100 dataset for 100 epochs. The other training hyperparameters are the same as those in implementation details for EfficientNet-B0 based models. 100 models are sampled in total and it takes about 2.5 GPU hours on average to train one model. The results of all the models are shown in Figure 1. Larger FLOPs lead to higher accuracy generally. Some of the sampled models perform better than the shrunken models using inversed giant formula of EfficientNet. For example, a sampled model with 318M FLOPs achieves 79.7% accuracy while EfficientNet-B0 with 387M FLOPs only achieves 78.8%. These observations indicate the necessity to design a more effective model shrinking method.

Table 1: TinyNet Performance on ImageNet-100. All the models are shrunken from the EfficientNet-B0 baseline. [†]Shrinking B0 to the minimum depth results in 173M FLOPs ($>$100M).

| Model | FLOPs | Acc. | Model | FLOPs | Acc. |
|---|---|---|---|---|---|
| EfficientNet-B$^{-1}$ | 200M | 75.8% | EfficientNet-B$^{-2}$ | 97M | 72.1% |
| shrink B0 by $r = 0.70$ | 196M | 74.9% | shrink B0 by $r = 0.46$ | 103M | 70.3% |
| shrink B0 by $d = 0.45$ | 196M | 76.5% | depth underflow[†] | - | - |
| shrink B0 by $w = 0.65$ | 205M | 77.2% | shrink B0 by $w = 0.38$ | 99M | 73.2% |
| TinyNet-B (ours) | 201M | **77.6**% | TinyNet-C (ours) | 97M | **74.1**% |

**Comparison to EfficientNet Rule.** In order to verify the effectiveness of the proposed model shrinking method, we compare our method with the inversed giant formula of EfficientNet and separately changing $r$, $d$ or $w$. From Table 1, the proposed method outperforms both EfficientNet rule and separately adjusting resolution, depth or width, demonstrating the effectiveness of the proposed tiny formula for model shrinking.

**Shrinking ResNet.** In addition to EfficientNet-B0, we also apply our method for shrinking the widely-used ResNet network architecture. ResNet-50 is adopted as the baseline model, and it is shrunken in different ways including reducing layers, EfficientNet rule and our method. The results on ImageNet-100 are shown in Table 2. Our models outperform other models generally, suggesting the effectiveness of the proposed model shrinking method for ResNet architecture.

Table 2: Performance of shrunken ResNet on ImageNet-100 dataset.

| Model | FLOPs | Acc. |
|---|---|---|
| Baseline ResNet-50 [12] | 4.1B | 78.7% |
| ResNet-34 [12] | 3.7B | 77.9% |
| Shrunken by EfficientNet rule | 3.7B | 78.3% |
| ResNet-50-A (ours) | 3.6B | **79.3**% |
| ResNet-18 [12] | 1.8B | 76.5% |
| Shrunken by EfficientNet rule | 1.8B | 76.9% |
| ResNet-50-B (ours) | 1.8B | **78.2**% |

## 4.3 Experiments on ImageNet-1000

The tiny formula obtained on ImageNet-100 can be well transferred to other datasets as demonstrated in NAS literature [61, 27, 49]. We evaluate our tiny formula on the large-scale ImageNet-1000 dataset to verify its generalization.

**TinyNet Performance.** We compare TinyNet models with other competitive small neural networks, including the models from original EfficientNet rule, *i.e.* EfficientNet-B$^{-\phi}$, and other state-of-the-art small CNNs such as MobileNet series [15, 38, 14], ShuffleNet series [58, 31], and MnasNet [42], are compared here. Several competitive NAS-based models are also included. Table 3 shows the performance of all the compared models. Our TinyNet models generally outperform other CNNs. In particular, our TinyNet-E achieves 59.9% Top-1 accuracy with 24M FLOPs, being 1.9% higher than the previous best MobileNetV3 Small 0.5$\times$ [14] with similar computational cost.

RandAugment [2] is a practical automated data augmentation strategy to improve the generalization of deep learning models. We use RandAugment with magnitude 9 and standard deviation 0.5 to improve the performance of our TinyNet-A and EfficientNet-B0, and show the results in Table 3. For the TinyNet models, RandAugment is beneficial to the performance. In particular, TinyNet-A + RA achieves 77.7% Top-1 accuracy which is 0.9% higher than vanilla TinyNet-A.

Table 3: Comparison of state-of-the-art small networks over classification accuracy, the number of weights and FLOPs on ImageNet-1000 dataset. "-" mean no reported results available.

| Model | Weights | FLOPs | Top-1 Acc. | Top-5 Acc. |
|---|---|---|---|---|
| MobileNetV3 Large 1.25× [14] | 7.5M | 356M | 76.6% | - |
| MnasNet-A1 [42] | 3.9M | 312M | 75.2% | 92.5% |
| Baseline EfficientNet-B0 [43] | 5.3M | 387M | 76.7% | 93.2% |
| TinyNet-A | 6.2M | 339M | **76.8%** | **93.3%** |
| EfficientNet-B0 [43] + RA | 5.3M | 387M | **77.7%** | **93.5%** |
| TinyNet-A + RA | 6.2M | 339M | **77.7%** | **93.5%** |
| MobileNetV2 1.0× [38] | 3.5M | 300M | 71.8% | 91.0% |
| ShuffleNetV2 1.5× [31] | 3.5M | 299M | 72.6% | 90.6% |
| FBNet-B [49] | 4.5M | 295M | 74.1% | - |
| ProxylessNAS [1] | 4.1M | 320M | 74.6% | 92.2% |
| EfficientNet-B$^{-1}$ | 3.6M | 201M | 74.7% | 92.1% |
| TinyNet-B | 3.7M | 202M | **75.0%** | **92.2%** |
| MobileNetV1 0.5× ($r$=0.86) [15] | 1.3M | 110M | 61.7% | 83.6% |
| MobileNetV2 0.5× [38] | 2.0M | 97M | 65.4% | 86.4% |
| MobileNetV3 Small 1.25× [14] | 3.6M | 91M | 70.4% | - |
| EfficientNet-B$^{-2}$ | 3.0M | 98M | 70.5% | 89.5% |
| TinyNet-C | 2.5M | 100M | **71.2%** | **89.7%** |
| MobileNetV2 0.35× [38] | 1.7M | 59M | 60.3% | 82.9% |
| ShuffleNetV2 0.5× [31] | 1.4M | 41M | 61.1% | 82.6% |
| MnasNet-A1 0.35× [42] | 1.7M | 63M | 64.1% | 85.1% |
| MobileNetV3 Small 0.75× [14] | 2.4M | 44M | 65.4% | - |
| EfficientNet-B$^{-3}$ | 2.0M | 51M | 65.0% | 85.2% |
| TinyNet-D | 2.3M | 52M | **67.0%** | **87.1%** |
| MobileNetV2 0.35× ($r$=0.71) [38] | 1.7M | 30M | 55.7% | 79.1% |
| MnasNet-A1 0.35× ($r$=0.57) [42] | 1.7M | 22M | 54.8% | 78.1% |
| MobileNetV3 Small 0.5× [14] | 1.6M | 23M | 58.0% | - |
| MobileNetV3 Small 1.0× ($r$=0.57) [14] | 2.5M | 20M | 57.3% | - |
| EfficientNet-B$^{-4}$ | 1.3M | 24M | 56.7% | 79.8% |
| TinyNet-E | 2.0M | 24M | **59.9%** | **81.8%** |

**Visualization of Learning Curves.** To better demonstrate the effect of our method, we plot the learning curves of EfficientNet-B$^{-4}$ and our TinyNet-E in Figure 4. From Figure 4(a), the accuracy of TinyNet-E is higher than that of EfficientNet-B$^{-4}$ by a large margin consistently during training. In the end of training, our TinyNet-E outperforms EfficientNet-B$^{-4}$ by an accuracy gain of 3.2%. The train and validation loss curves in Figure 4(b) also show the superiority of our TinyNet.

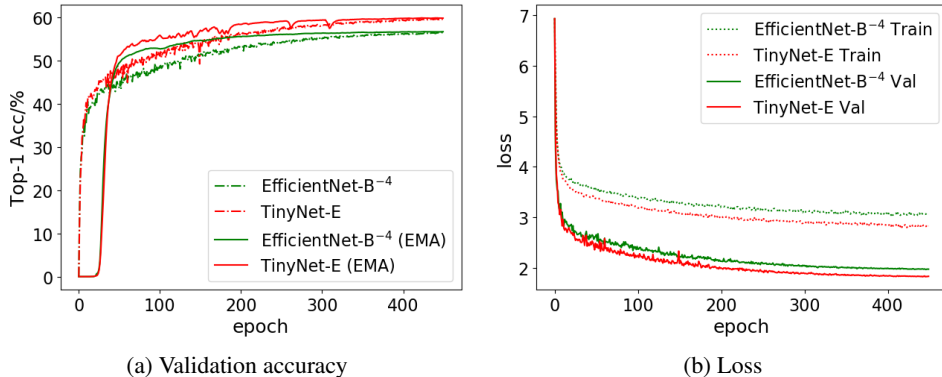

(a) Validation accuracy  (b) Loss

Figure 4: Learning curves of EfficientNet-B$^{-4}$ and our TinyNet-E on ImageNet-1000.

**Visualization of Class Activation Map.** We visualize the class activation map [59] for EfficientNet-B$^{-4}$ and TinyNet-E to better demonstrate the superiority of our TinyNet. The images are randomly picked from ImageNet-1000 validation set. As shown in Figure 5, TinyNet-E pays attention to the more relevant regions, while EfficientNet-B$^{-4}$ sometimes only focuses on the unrelated objects or the local part of target objects.

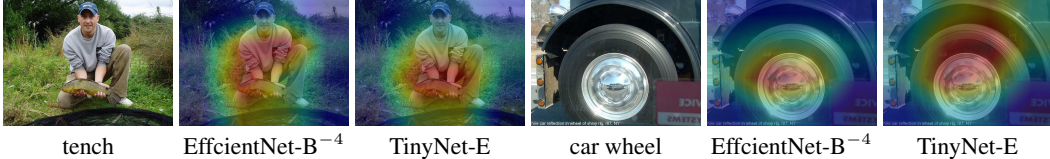

| tench | EffcientNet-B$^{-4}$ | TinyNet-E | car wheel | EffcientNet-B$^{-4}$ | TinyNet-E |

Figure 5: Class Activation Map for EfficientNet-B$^{-4}$ and our TinyNet-E.

**Shrinking by $r, d, w$ Separately.** We also compare the proposed model shrinking rule with the naive method, *i.e.* changing resolution, depth and width separately. We tune resolution, width or depth separately to form models with 100M and 200M FLOPs. Note that the minimum viable depth for EfficientNet-B0 is reached with 7 inverted residual bottlenecks, and the corresponding FLOPs 174M. The results are shown in Figure 6. In general, all shrinking approaches lead to lower accuracy when the number of FLOPs decreases, but our model shrinking method can alleviate the accuracy drop, suggesting the effectiveness of the proposed method.

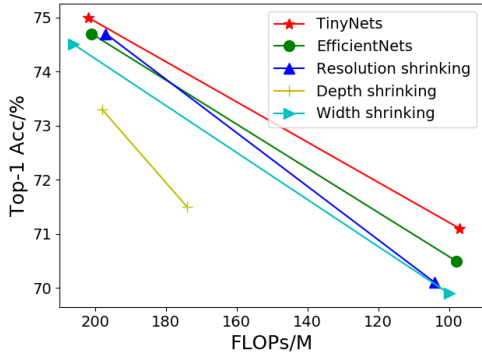

Figure 6: Accuracy *v.s.* FLOPs of the models shrunken using different ways on ImageNet-1000.

**Inference Latency Comparison.** We also measure the inference latency of several representative CNNs on Huawei P40 smartphone. We test under single-threaded mode with batch size 1 [43, 14] using the MindSpore Lite tool [16]. The results are listed in Table 4, where we run 1000 times and report average latency. Our TinyNet-A runs 15% faster than EfficientNet-B0 while their accuracies are similar. TinyNet-E can obtain 3.2% accuracy gain compared to EfficientNet-B$^{-4}$ with similar latency.

Table 4: Inference latency comparison.

| Model | FLOPs | Latency | Top-1 | Model | FLOPs | Latency | Top-1 |
|---|---|---|---|---|---|---|---|
| EfficientNet-B0 | 387M | 99.85 ms | 76.7% | EfficientNet-B$^{-4}$ | 24M | 11.54 ms | 56.7% |
| TinyNet-A | 339M | 81.30 ms | 76.8% | TinyNet-E | 24M | 9.18 ms | 59.9% |

**Generalization on Object Detection.** To verify the generalization of our models, we apply tinynets on object detection task. We adopt SSDLite [28, 38] with $512 \times 512$ input as baseline network due to its efficiency and test on MS COCO dataset [26]. The experimental setting is similar to that in [38]. From results in Table 5, we can see that our TinyNet-D outperforms EfficientNet-B$^{-3}$ by a large margin with comparable computational cost.

Table 5: Results on MS COCO dataset.

| Model | Backbone FLOPs | mAP | AP$_{50}$ | AP$_{75}$ | AP$_S$ | AP$_M$ | AP$_L$ |
|---|---|---|---|---|---|---|---|
| EfficientNet-B$^{-3}$ | 51M | 17.1 | 29.9 | 17.0 | 5.2 | 30.5 | 54.8 |
| TinyNet-D | 52M | 19.2 | 32.6 | 19.2 | 7.0 | 33.9 | 57.5 |

# 5   Conclusion and Discussion

In this paper, we study the model Rubik's cube for shrinking deep neural networks. Based on a series of observations, we find that the original giant formula in EfficientNet is unsuitable for generating smaller neural architectures. To this end, we thoroughly analyze the importance of resolution, depth and width w.r.t. the performance of portable deep networks. Then, we suggest to pay more concentration on the resolution and depth and calculate the model width to satisfy FLOPs constraint. We explore a series of TinyNets by utilizing the tiny formula to twist the three dimensions. The experimental results for both EfficientNets and ResNets demonstrate effectiveness of the proposed simple but effective scheme for designing tiny networks. Moreover, the tiny formula in this work is summarized according to the observation on smaller models. These smaller models can also be further enlarged to obtain higher performance with some new rules beyond the giant formula in EfficientNets, which will be investigated in future works.

## Broader Impact

The widely usage of deep neural networks which require large amount of computation resource is putting pressure on the energy source and the natural environment. The proposed model shrinking method for obtaining tiny neural networks is beneficial to energy conservation and environment protection.

## Funding Disclosure

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
