[Supplementary Material]

# Model Rubik's Cube: Twisting Resolution, Depth and Width for TinyNets

**Kai Han**[1,2*] **Yunhe Wang**[1*] **Qiulin Zhang**[1,3] **Wei Zhang**[1] **Chunjing Xu**[1] **Tong Zhang**[4]
[1]Noah's Ark Lab, Huawei Technologies
[2]State Key Lab of Computer Science, ISCAS & UCAS
[3]BUPT    [4]HKUST

## A    Appendix

### A.1    Experiments on GhostNet

We also verify the effectiveness of the proposed model Rubik's cube on the state-of-the-art portal network GhostNet [3]. We first build a baseline GhostNet with about 591M FLOPs, which is denoted as GhostNet-A (details in Table 2). We start from GhostNet-A and shrink the model by the proposed tiny formula, resulting in a serious of smaller models, *i.e.* GhostNet-B/C/D. The new models are trained using the similar setting as TinyNets. The comparison with the original GhostNets on ImageNet dataset is shown in Table 1. We can see that the new GhostNet models obtained by the proposed model Rubik's cube outperform the original GhostNets which only change the width.

Note that GhostNets use ReLU as activation function, and more complex activations (*e.g.* HSwish [4]) may further improve the performance. We also show the results with automated data augmentation strategy, *i.e.* RandAugment [2] in Table 1. Both fancy activation function and data augmentation could boost GhostNets to higher performance.

For better comparison, we plot the results in Fig. 1. The compared methods include recent state-of-the-art models, *i.e.* MobileNetV2 [6], MobileNetV3 [4], EfficientNet [8], ShuffleNetV2 [5], FBNet [9], MnasNet [7], ProxylessNAS [1] and GreedyNAS [10]. GhostNet models enhanced by TinyNet technique consistently outperform other models by a significant margin.

Figure 1: Top-1 Accuracy *v.s.* FLOPs on ImageNet dataset.

---

Table 1: GhostNet results on ImageNet dataset.

| Model | Weights | FLOPs | Top-1 Acc. | Top-5 Acc. |
|---|---|---|---|---|
| EfficientNet-B1 [8] | 7.8M | 700M | 78.7% | - |
| GhostNet-A | 11.9M | 591M | 78.7% | 94.2% |
| GhostNet-A + HSwish | 11.9M | 591M | 78.9% | 94.4% |
| GhostNet-A + HSwish + RA | 11.9M | 591M | 79.4% | 94.5% |
| EfficientNet-B0 [8] | 5.3M | 387M | 76.7% | 93.2% |
| GhostNet-1.5× [3] | 9.1M | 300M | 76.4% | 92.9% |
| GhostNet-B | 8.0M | 300M | 77.0% | 93.3% |
| GhostNet-B + HSwish | 8.0M | 300M | 77.1% | 93.4% |
| GhostNet-B + HSwish + RA | 8.0M | 300M | 77.6% | 93.5% |
| MobileNetV3 Large 0.75× [4] | 4.0M | 155M | 73.3% | - |
| GhostNet-1.0× [3] | 5.2M | 141M | 73.9% | 91.4% |
| GhostNet-C | 5.0M | 141M | 74.2% | 91.7% |
| GhostNet-C + HSwish | 5.0M | 141M | 74.8% | 92.0% |
| GhostNet-C + HSwish + RA | 5.0M | 141M | 75.0% | 92.0% |
| MobileNetV3 Small 0.75× [4] | 2.4M | 44M | 65.4% | - |
| GhostNet-0.5× [3] | 2.6M | 42M | 66.2% | 86.6% |
| GhostNet-D | 2.6M | 52M | 67.7% | 87.5% |
| GhostNet-D + HSwish | 2.6M | 52M | 68.4% | 87.8% |
| GhostNet-D + HSwish + RA | 2.6M | 52M | 68.6% | 88.1% |

Table 2: GhostNet-A architecture. #exp means expansion ratio. #out means the number of output channels. SE denotes whether using SE module (reduction ratio 10). #repeat denotes repeat times.

| Input size | Operator | #exp | #out | SE | Stride | #repeat |
|---|---|---|---|---|---|---|
| $240^2 \times 3$ | Conv 3×3 | - | 28 | - | 2 | 1 |
| $120^2 \times 28$ | G-bneck | 1 | 28 | 1 | 1 | 1 |
| $120^2 \times 28$ | G-bneck | 3 | 44 | 1 | 2 | 1 |
| $60^2 \times 44$ | G-bneck | 3 | 44 | 1 | 1 | 1 |
| $60^2 \times 44$ | G-bneck | 3 | 72 | 1 | 2 | 1 |
| $30^2 \times 72$ | G-bneck | 3 | 72 | 1 | 1 | 2 |
| $30^2 \times 72$ | G-bneck | 6 | 140 | 1 | 2 | 1 |
| $15^2 \times 140$ | G-bneck | 2.5 | 140 | 1 | 1 | 3 |
| $15^2 \times 140$ | G-bneck | 6 | 196 | 1 | 1 | 3 |
| $15^2 \times 196$ | G-bneck | 6 | 280 | 1 | 2 | 1 |
| $8^2 \times 280$ | G-bneck | 6 | 280 | 1 | 1 | 5 |
| $8^2 \times 280$ | Conv 1×1 | - | 1680 | - | 1 | 1 |
| $8^2 \times 1680$ | Pooling | - | - | - | - | 1 |
| $1^2 \times 1680$ | Conv 1×1 | - | 1400 | - | 1 | 1 |
| $1^2 \times 1400$ | FC | - | 1000 | - | - | 1 |