[Reviews · NeurIPS 2020]

Review 1

Summary and Contributions: In contrast with EfficientNet that scales up a base network, this paper proposes an inverse formula to scale down the model in terms of resolution, depth, and width. The effectiveness of the tiny formula is evaluated extensively on ImageNet-100 and ImageNet-1000 classification benchmarks.

Strengths: * The tiny formula seems to work well empirically. The compact models it produces outperform the EfficientNet scaling rule consistently over multiple model capacities. * The paper is overall well-written, with detailed tables and informative figures. Experiment section and ablation studies are extensive. * Hyperparameter settings and training details are included, which should make the results sufficiently reproducible. * I appreciate the report of runtime statistics on real hardware (Table 4, inference latency with TFLite framework). This is much more useful than proxy metrics like FLOP.

Weaknesses: * While the tiny formula works well empirically, it is a straightforward extension of the compound scaling idea from EfficientNet. The method definitely has novelty, but not highly significant. * The accuracy of EfficientNet-B^{-1} seems to be wrong (likely a typo). According to Figure 2, it should be around 76%, but the text says 78.5%. * The curves in Figure 6 only have two end points, which make them less informative. It'd be much better to see the performances of these shrinking strategies across a range of FLOPs. EDIT AFTER REBUTTAL ================== I have read the authors' rebuttal and other reviews. I agree with the negative reviews that this paper may lack significant novelty, but I still think that it has value to the community as a shrinking extension of EfficientNet. In addition, the authors addressed my concerns very well in the rebuttal. My updated position is weak accept.

Correctness: Yes

Clarity: Overall, it is quite well-written.

Relation to Prior Work: Yes

Reproducibility: Yes

Additional Feedback:


Review 2

Summary and Contributions: This paper explored the twisting rules for smaller neural networks and found that resolution and depth are more important than width for tiny networks. Based on this observation, a tiny formula for downsizing neural architectures was proposed and the resulted tiny networks showed better performance than the famous EfficientNet and MobileNetV3.

Strengths: 1. This paper aimed to explore twisting resolution, depth and width for obtaining smaller neural networks, which was inverse to the EfficientNet for bigger networks. This paper evaluated the importance of resolution, depth and width empirically, and found original EfficientNet rule lose its efficiency for smaller models. The authors further found that resolution and depth are more important than width for tiny networks. The empirical evaluation sounds reasonable and the findings are new to the machine learning community. 2. Based on the above observations, the paper further proposed the tiny formula for extremely small neural networks. Given the target FLOPs, the formula outputs the resolution, depth and width of the best models, respectively. The nonparametric Gaussian process regression was used to instantiate the formula based on the empirical evaluation on a number of randomly sampled neural architectures. 3. The resulted tiny networks showed better performance than the baseline and other twisting rules. Especially, the Tinynet series can obtain state-of-the-art performance on the large-scale ImageNet dataset in a mobile setting (e.g. FLOPS<600M).

Weaknesses: 1. Why using Gaussian process regression for designing the tiny formula in Page 4. It is better to compare it with other methods such as conventional linear regression? 2. The authors should discuss the relationship between the proposed Model Rubik’s Cube and Neural Architecture Search (NAS) methods, which can also be used to search better resolution, depth and width? 3. What’s the detailed setting of RandAugment in Page 8?

Correctness: Claims and method seem OK to me.

Clarity: The paper is well written and easy to follow. Several typos: Spearman correlation coefficient (Spearmanr) are calculated --> Spearman correlation coefficient (Spearmanr) is calculated, etc.

Relation to Prior Work: This work has revisited the related works including model compression and resolution/depth/width of CNNs. The difference with giant formula in EfficientNet is discussed where the inversed giant formula cannot be well applied to shrink neural networks so the new tiny formula is necessary to explore for obtaining smaller neural networks with high performance.

Reproducibility: Yes

Additional Feedback: ========post rebuttal ============ I have read the rebuttal and reviews from other reviewers. My concerns are well addressed. But I like to mention that some rebuttal text or results need to be put in the main text if there is still some possible space.


Review 3

Summary and Contributions: This paper are focusing on exploring efficient deep neural networks with fewer model parameters and computational costs from resolution, depth and width dimensions. They observed that resolution and depth are more important than width for compact models which is different from compound scaling in EfficientNet. The proposed model achieved about 1.9% higher than that of the previous MobileNetV3 with similar computational cost. 


Strengths: 1. I think the discovery that “resolution and depth are more important than width for compact models” is interesting and can somewhat benefit to the community. 2. This paper achieved competitive result — about 1.9% higher than that of MobileNetV3 with a similar computational cost. 3. The proposed method is simple and easy to follow.

Weaknesses: 1. This paper still considers the only resolution, depth and width dimensions, which have been studied in EfficientNet. Although the discovery in this paper that “resolution and depth are more important than width for tiny networks” is different from the conclusion in EfficientNet, I feel this point is not significant enough and it seems like just a supplement for EfficientNet. 2. The proposed method in this paper is fairly heuristic, the authors trained a mass of models then observed the phenomenon that resolution and depth are more important. I’m not saying that this kind of method is not good, but I think the insights and intuitions why resolution and depth are more important than width for small networks (derived from this way) are still not clear. 3. In my opinion, this paper is basically doing random search by shrinking the EfficientNet-B0 structure configurations on the mentioned three dimensions, I believe the derived observation is useful but the method itself contains very limited value to the community. Even some simple searching method like evolutionary searching can achieve similar or the same purpose through a more efficient way. 4. I’m a little bit confused about the class activation map in Fig. 4. The authors claimed their model can focus on more relevant regions, while EfficientNet-B−4 only focuses on the unrelated objects or the local part of target objects. In their method, I did not see any part/module that has this function or can enhance the ability to let network focus on the more relevant regions. Do these visualizations mean that models with better performance can focus on the more relevant regions. So I’m wondering whether these images are really randomly picked or just cherry-picked. 5. How about the total training cost, and the comparisons with other searching methods, as I feel the proposed method seems somewhat inefficient. ================= Post rebuttal ================= After reading the authors' rebuttal and other reviewers' comments, I still think the method proposed in this paper is too heuristic and not efficient. FBNetV2 also considers the resolution, depth and width for the tiny nets, but has higher accuracy and more efficient searching process. I also have concerns about the experiments, it seems like the baseline EfficientNet-B0 is not as good as the reported accuracy in the official paper and even inconsistent to the github results.

Correctness: I think the claims, method and empirical methodology in this paper are correct, but some visualizations are not clear to me. Also the authors mentioned that "The EfficientNet-B0 based models are trained using similar settings as [33]". It seems that they did not use exactly the same hyper-parameters to train EfficientNet-B0, I'm not sure if the difference will affect the performance or not.

Clarity: Generally, the paper is well written and the description is clear.

Relation to Prior Work: Yes.

Reproducibility: Yes

Additional Feedback:


Review 4

Summary and Contributions: The paper introduces a method that establishes a relationship between model width, model depth, and input resolution. The results under very small constraint budgets (< 25 MFLOPs) are interesting.

Strengths: i) In contrast to manually designed relationship between width, depth and resolution, this paper formulates the relationship between these variables using gaussian process. ii) Results under small computational constraints are interesting. iii) Augmentation plays a key role in the performance of CNNs. The insight about w/ and w/o auto-augmentations under different FLOPs are interesting and would help researchers in future to carefully choose the augmentation methods.

Weaknesses: i) The gains with respect to EfficientNet scaling are marginal in most FLOP settings (except 20-25M setting) and within noise levels. This raises concerns about the scalability of the proposed approach. ii) The compound scaling method introduced in EfficientNet works across different models and tasks, showing the method is generic. A study on different tasks (e.g., object detection and semantic segmentation) is required to understand the true benefits of the proposed method along with one or two experiments on large models. iii) Model width scaling is widely studied in manually designed networks (eg., ESPNetv2, ShuffleNetv1-v2, MobileNets, etc.). Discussion about these methods is missing in related work (though authors cite some of them in experimental section). iv) Is the approach generalizable? Can it be applied to directly to NLP model, for example transformers? A discussion about generalizability is missing. Post rebuttal =============== I have read the author response and my opinion remains the same. I am still not fully convinced by authors response. (1) Contributions of MobileNetv2 and ShuffleNetv1 are more on architectural design and not on outperforming SOTA methods, so argument based on these comparisons are not correct. The ImageNet dataset is noisy as several previous studies have shown (e.g., Label-refinery) and I am not sure what 0.5% gain in accuracy means on this task. I am glad that authors included results on MS-COCO and I can see that proposed approach is effective. However, more studies are required especially using SSD as a backbone architecture because (1) that is widely used in mobile settings and (2) to see the improvements over SOTA networks that works in those settings. With authors response, I am not able to evaluate the merit of the proposed method with respect to existing efficient models (e.g., MobileNetv3 and MixNet) that are aimed for mobile devices. Though I like the paper, I am not convinced by the experiments. To improve the paper, I would suggest authors to include: (1) Comparison with EfficientNet on at least on one of the data points that is officially reported. Like EfficientNet, do a grid search to find a scaling parameter for small and medium-size models and then compare the results. (2) I think MobileNetv3 is the most appropriate baseline network for this paper. This is because (1) it uses the same basic building block as EfficientNet and (2) it is designed for mobile devices. Comparison on MobileNetv3 with and without scaling would definitely strengthen the proposed method. (3) Use SSD for object detection rather than Faster-RCNN, because it is widely used in efficient networks for mobile devices. Unlike ROIAlignLayer, it does not pose memory alignment issues.

Correctness: Yes

Clarity: The paper is well-written and easy to follow.

Relation to Prior Work: A discussion about manually designed efficient and light-weight networks is required (though authors cite few of them in results section).

Reproducibility: Yes

Additional Feedback:

[Author Response · NeurIPS 2020]

1 We sincerely thank all the reviewers for their helpful comments.

2 **Response to Reviewer #1:**

3 **Q:** *Relationship with EfficientNet.* **A:** Our main contributions: 1) Our
method provides a general pipeline for shrinking neural networks, in con-
trast to EfficientNet that enlarges CNNs. 2) We found that "resolution and
depth are more important than width for tiny networks" which is different
from EfficientNet. 3) Different to the giant formula in EfficientNet that is
handcrafted, the proposed data-driven formula twists the three dimensions
based on the observation of frontier models. 4) The obtained TinyNets
show outstanding performance.

12 **Q:** *Typo.* **A:** The actual accuracy is 75.8%, not 78.5%. We'll fix it.

13 **Q:** *More FLOPs in Fig.6.* **A:** We include more FLOPs as shown in Fig⇒

14 **Response to Reviewer #2:**

15 **Q:** *Why Gaussian process regression.* **A:** In Fig.3, the curves are not simple linear regression and unknown to us. We
utilize Gaussian process regression to fit the curves for its effectiveness and nonparametric property.

17 **Q:** *Relationship with NAS.* **A:** The main differences: 1) Our method gives principle to obtain smaller models which
can be interpretable and general, while NAS only results in several architectures which are hard to understand. 2) The
formula in our method can output the configurations for any FLOPs, while NAS only search for predefined FLOPs.

20 **Q:** *Setting of RandAugment.* **A:** We set the magnitude of RandAugment to be 9 with a std as 0.5 in all networks.

21 **Response to Reviewer #3:**

22 **Q:** *Just a supplement for EfficientNet.* **A:** Thanks for this comment. In fact, resolution, depth and width are three key
dimensionalities for the performance and computational cost of neural networks. This paper aims to transfer the success
of EfficientNets for enlarging models to tiny models with higher performance. We found that "resolution and depth are
more important than width for tiny networks" which is different from EfficientNet. Besides, we present a data-driven
algorithm for finding the optimal tiny formula by twisting the three dimensions based on the observation of frontier
models. The performance of TinyNets is about 0.3-3.8% higher than that of EfficientNet with similar computational
complexity, which is a significant improvement on ImageNet.

29 **Q:** *Why resolution & depth > width.* **A:** Basically, The resolution contain the raw information of images, *e.g.* the average
resolution of ImageNet images is 482x418 pixels, so changing too much of resolution will cause severe information loss
to the objects. The depth directly controls the times of nonlinear transformation and the receptive field of CNNs, which
is important for extracting hierarchical features from images. As for width, the channels contains more redundancy
as shown in many pruning methods, so the width is not so important as resolution and depth. More discussions and
visualizations will be included in the final version.

35 **Q:** *Evolutionary searching.* **A:** Thanks for this nice concern. Random sampling here provides a simple yet effective
approach for TinyNets with higher performance and compact architectures. We also agree that the evolutionary
algorithm can provide better results which can be explored as an extension of our method.

38 **Q:** *CAM in Fig.4.* **A:** We follow the EfficientNet paper and show these class activation map figures. These visualizations
only mean that models with better performance can focus on the more relevant regions.

40 **Q:** *Training cost.* **A:** Different from NAS which gives several 'strange' architectures, our method provides general and
interpretable formula for shrinking CNNs. We randomly sample 100 models and train them on a ImageNet subset with
cost of 10.4 GPUdays, compared to 3800 GPUdays@MnasNet, 8.3 GPUdays@ProxylessNAS, 9 GPUdays@FBNet.

43 **Q:** *Hyper-parameters to train EfficientNet.* **A:** We use the thirdparty Pytorch code which reproduces the official
Tensorflow code with similar performance (diff≤0.1%). All the TinyNets and EfficientNets are trained under the same
settings, so the results and conclusion are fairly given.

46 **Response to Reviewer #4:**

47 **Q:** *The gains are marginal.* **A:** Thanks for this nice concern. We have hilighted the accuracy gains of our TinyNets
with similar FLOPs to those of EfficientNets in Table R1. Considering that the ImageNet is still a difficult benchmark
for modern lightweight CNNs, these gains are exactly significant improvements. For example, MobileNetV2 is about
0.5% higher than that of ShuffleNetV1 (SOTA of that time).

51 **Q:** *A study on different tasks.* **A:** The comparsion experiments on COCO dataset as shown in Table R2 illustrate that our
TinyNet can also surpass the conventional EfficientNet on the Faster RCNN framework.

53 **Q:** *Related work.* **A:** We will cite and discuss about the width scaling methods used in these recent excellent networks
including ESPNetv2, ShuffleNetv1-v2, MobileNets.

55 **Q:** *Generalizability.* **A:** Thanks for this constructive comment. This paper is mainly focusing on tiny models for
computer vision tasks. To apply our method to NLP models, some topics need to be explored accordingly, *e.g.* how to
define 'resolution' in NLP models. More discussion on future work and broader impact will be included.

Table R1. ImageNet gains of TinyNet over EfficientNet.

| FLOPs/M | 339 | 202 | 97 | 52 | 24 |
|---|---|---|---|---|---|
| Acc gain/% | 0.5 | 0.3 | 0.6 | 1.9 | 3.8 |

Table R2. COCO performance of TinyNet & EfficientNet.

| | | |
|---|---|---|
| EfficientNet (Backbone FLOPs/mAP) | 387M/30.4% | 98M/18.5% |
| TinyNet (Backbone FLOPs/mAP) | 339M/30.5% | 97M/19.8% |

[Meta-Review · NeurIPS 2020]

The paper received mixed ratings: two reviewers recommend acceptance, and two reviewers consider the paper is marginally below the threshold. All reviewers agree that the paper provides useful insights, e.g., the observation that resolution and depth are more important than width for tiny networks. The main concerns raised by the reviewers were (i) novelty is not highly significant/the method is too heuristic (ii) issues with experiments and lack of analysis on other tasks, such as object detection. The rebuttal helped clarify several other questions raised by the reviewers, and included new experiments on COCO object detection using Faster-RCNN. All reviewers actively participated in the discussion phase. R3 remained concern about the search efficiency of the method with respect to other alternatives such as FBNetv2, and pointed out issues with the reported results for EfficientNetB0 (Table 3). R4 remained concerned about the generalization of the approach for detection when faster methods such as SSD or YOLO are used. While the concerns raised by R3 and R4 are legitimate, the AC (after a discussion with SAC and another AC) agrees with R1 and R2 that the paper passes the acceptance bar of NeurIPS. The inverse formula to scale down EfficientNet has value to the community. The results are strong, especially when considering low flops (despite the fact flops may not be a good proxy for actual speed). It would be desirable to see the performance of the method in tandem with efficient detectors such as YOLO or SSD, but the AC considers the experimental analysis in the paper is sufficient. The authors should clarify the reported EfficientNet B0 results in the final version, but this is not a major issue given the results on the low flop regime. Please also make sure to add the discussion in the rebuttal to the camera-ready version.